# Study on Hyperspectral Monitoring Model of Total Flavonoids and Total Phenols in Tartary Buckwheat Grains

**DOI:** 10.3390/foods12071354

**Published:** 2023-03-23

**Authors:** Chenbo Yang, Lifang Song, Kunxi Wei, Chunrui Gao, Danli Wang, Meichen Feng, Meijun Zhang, Chao Wang, Lujie Xiao, Wude Yang, Xiaoyan Song

**Affiliations:** College of Agriculture, Shanxi Agricultural University, Taigu, Jinzhong 030801, China

**Keywords:** tartary buckwheat, hyperspectral, total flavonoids, total phenols, preprocessing

## Abstract

Tartary buckwheat is a common functional food. Its grains are rich in flavonoids and phenols. The rapid measurement of flavonoids and phenols in buckwheat grains is of great significance in promoting the development of the buckwheat industry. This study, based on multiple scattering correction (MSC), standardized normal variate (SNV), reciprocal logarithm (Lg), first-order derivative (FD), second-order derivative (SD), and fractional-order derivative (FOD) preprocessing spectra, constructed hyperspectral monitoring models of total flavonoids content and total phenols content in tartary buckwheat grains. The results showed that SNV, Lg, FD, SD, and FOD preprocessing had different effects on the original spectral reflectance and that FOD can also reflect the change process from the original spectrum to the integer-order derivative spectrum. Compared with the original spectrum, MSC, SNV, Lg, FD, and SD transformation spectra can improve the correlation between spectral data and total flavonoids and total phenols in varying degrees, while the correlation between FOD spectra of different orders and total flavonoids and total phenols in grains was different. The monitoring models of total flavonoids and total phenols in grains based on MSC, SNV, Lg, FD, and SD transformation spectra achieved the best accuracy under SD and FD transformation, respectively. Therefore, this study further constructed monitoring models of total flavonoids and total phenols content in grains based on the FOD spectrum and achieved the best accuracy under 1.6 and 0.6 order derivative preprocessing, respectively. The R^2^_c_, RMSE_c_, R^2^_v_, RMSE_v_, and RPD were 0.8731, 0.1332, 0.8384, 0.1448, and 2.4475 for the total flavonoids model, and 0.8296, 0.2025, 0.6535, 0.1740, and 1.6713 for the total phenols model. The model can realize the rapid measurement of total flavonoids content and total phenols content in tartary buckwheat grains, respectively.

## 1. Introduction

Buckwheat is a common food crop. As a functional food rich in flavonoids, phenols, and other functional components, buckwheat has attracted more and more attention [1,2,3,4]. At present, studies have shown that flavonoids and phenols play an important role in the treatment of diabetes, hypertension, and other diseases, and in improving the antioxidant effect of the body [5,6,7]. Accurately measuring the content of flavonoids and phenols is the premise of using them. At present, the common methods for the measurement of flavonoids mainly include liquid chromatography [8] and spectrophotometry [9]. The methods for the measurement of phenols mainly include chromatography [10], mass spectrometry [11], and spectrophotometry [12]. Although these methods have the advantages of strong repeatability and high accuracy to varying degrees, they also have the disadvantages of complex operation steps and of being time-consuming and laborious. Therefore, finding a method that can quickly measure these substances is of great significance to improving the utilization efficiency of flavonoids and phenols in buckwheat.

Hyperspectral remote sensing technology has become an important technical means for many researchers to analyze the target quantitatively and qualitatively because it can obtain the characteristics of the target quickly and losslessly [13,14,15]. Agriculture is one of the important application directions of remote sensing technology. Predecessors have used hyperspectral remote sensing technology to conduct a lot of research on soil organic carbon [16], nitrogen [17], and water content [18], crop biomass [19], chlorophyll [20,21], and leaf area index [22], as well as crop grain quality [23]. Some researchers also used spectral technology to study the flavonoids content in buckwheat grains. For example, Wang et al. [24] used near-infrared spectroscopy to monitor the content of total flavonoids in buckwheat grains and obtained a model with high accuracy, whose validation R^2^ reached 0.967. Rutin is a kind of flavonoid. Yang et al. [25] constructed a near-infrared spectral prediction model of rutin content in buckwheat grains by using the partial least squares regression (PLSR) method, with a prediction R^2^ of 0.76, which had a good prediction effect. Ladan et al. [26] quantitatively analyzed the rutin content of buckwheat grains of seven different varieties, and the correlation coefficient of the validation model reached 0.98. In general, it is feasible to use hyperspectral remote sensing technology to quantitatively monitor the content of flavonoids in buckwheat grains. At the same time, there are few reports on the hyperspectral monitoring of phenols in buckwheat grains.

Research showed that proper preprocessing of hyperspectral data before constructing a hyperspectral monitoring model is an important means to reduce spectral noise and improve the model accuracy [27]. For example, Xu et al. [28] conducted Svitzky–Golay filter (SGF), wavelet packet transform, multiple scattering correction (MSC), and derivatives transformation on spectral data before constructing their hyperspectral monitoring model of soil organic matter. The results showed that the denoising effect of SGF was higher than that of other methods and that low-order, fractional-order derivative (FOD) preprocessing can highlight the spectral characteristics of soil. It can be seen that it is very necessary to preprocess the hyperspectral data properly before constructing a hyperspectral monitoring model.

However, although the effect of derivative preprocessing on the model was also considered in different degrees when constructing the hyperspectral monitoring model of total flavonoids content, only the integer-order derivative (IOD) was considered, and the effect of FOD preprocessing on the model was not further considered when the IOD preprocessing had a good effect.

In this study, through selenium treatment of buckwheat, the total flavonoids, total phenols content, and hyperspectral reflectance of buckwheat grains were obtained. The hyperspectral reflectance data were preprocessed using MSC, standardized normal variate (SNV), reciprocal logarithm (Lg), first-order derivative (FD), and second-order derivative (SD). The models were constructed by PLSR. At the same time, we judged whether derivative preprocessing would be a better preprocessing algorithm. If the effect of FD or SD was better, further FOD preprocessing was carried out. This paper aims to: (1) study the effects of MSC, SNV, Lg, FD, and SD preprocessing on the accuracy of hyperspectral monitoring models of total flavonoids and total phenols in tartary buckwheat grains; (2) when the accuracy of the model based on FD or SD spectra is higher, further analyze whether the accuracy of the model based on the FOD spectrum is higher; and (3) realize the rapid acquisition of total flavonoids and total phenols content in tartary buckwheat grains.

## 2. Materials and Methods

### 2.1. Experimental Design

The experiment adopted an orthogonal experimental design. The experiment was conducted in Taigu County, Jinzhong City, Shanxi Province from June to October 2018, and Youyu County, Shuozhou City, Shanxi Province from May to September 2018. A total of 8 treatments of 0, 1.37, 2.74, 5.48, 8.22, 12.33, 18.495, and 27.7425 g·hm^−2^ were set, and each treatment was repeated three times. Among them, the tested varieties in Taigu County were “Heifeng 1 hao” and “Jinqiao 2 hao”. The branching stage and early flowering stage were selected for treatment, and the area of each plot was 12 m^2^ (3 m × 4 m), a total of 48 plots. The tested varieties in Youyu County were “Heifeng 1 hao” and “Jinqiao 6 hao”. The early flowering stage and peak flowering stage were selected for treatment, and the area of each plot was 10 m^2^ (2 m × 5 m), a total of 48 plots. Selenium was applied to soil and leaves. The specific orthogonal design scheme is shown in Table 1.

### 2.2. Index Measurement

After the tartary buckwheat was mature, the grains were harvested and dried indoors for seven days, so that the water content of all samples was basically at the same level. Then, the grains were shelled, crushed, and screened through a 0.2 mm sieve (Zhangxing Sieve, Shaoxing, China), so that all the crushed samples remained the same thickness [24]. The sieved samples were used to measure hyperspectral reflectance, total flavonoids, and total phenols content. We used the FieldSpec 3 portable hyperspectral radiometer (ASD Company, Boulder, CO, USA) to measure the hyperspectral reflectance of the grain powder. The collection band range was 350 nm–2500 nm. Before measurement, we placed the sample into a black petri dish and flattened it. Then, a plant probe was used to measure the spectrum; a whiteboard was used to correct before each measurement. The total flavonoids content was measured using colorimetry [29]. The total phenols content was measured using the Solarbio Kit [30].

### 2.3. Data Analysis

In this study, the effect of the preprocessing algorithm in the process of constructing the model was analyzed, preliminarily by using correlation analysis through different preprocessing methods of the original spectrum. Then, the PLSR method was used to construct the model, and the final model accuracy was used to evaluate the advantages and disadvantages of the preprocessing algorithm. The algorithm and model evaluation indicators used are as follows.

#### 2.3.1. Preprocessing Algorithm

MSC [31] can eliminate the scattering effect caused by uneven sample distribution. The specific calculation steps are as follows:

(1) Calculate the average spectral reflectance of all samples and record it as $\bar{x}$;

(2) The spectral reflectance *x_i_* of each sample is linearly regressed with $\bar{x}$ to obtain the baseline offset *b_i_* and baseline shift *c_i_*;

(3) Subtract *c_i_* from *x_i_* and divide by *b_i_* to obtain the corrected spectrum *x_msc_* of each sample.

The SNV [32] can eliminate the influence of diffuse reflection on the spectrum. The calculation equation is as follows (1):(1)$$x_{snv}=\frac{x-\bar{x}}{x_{std}}$$
where $\bar{x}$ and *x_std_* are the average value and the standard deviation of spectral reflectance of all samples, respectively.

Lg [33,34] transformation takes the reciprocal first and then logarithm of the original spectrum, which can reduce the influence of multiplicative factors caused by illumination changes.

Derivative transformation refers to the derivation of the original spectral curve, which can be divided into IOD and FOD. Among them, IOD is more used as FD and SD [35,36], while FOD can be used to calculate any real-order derivative [37]. The FD, SD, and FOD can be calculated by the following equations [37]:(2)$$f^{\prime }\left(x\right)=\underset{h\to 0}{\lim}\frac{f\left(x+h\right)-f\left(x\right)}{h}$$
(3)$$f^{\prime \prime }\left(x\right)=\underset{h\to 0}{\lim}\frac{f\left(x+2h\right)-2f\left(x+h\right)+f\left(x\right)}{h^{2}}$$
(4)$$f^{\alpha }\left(x\right)\approx \overset{t}{\underset{m=0}{\sum }}\frac{\Gamma \left(-\alpha +1\right)}{m!\;\Gamma \left(-\alpha +m+1\right)}f\left(\lambda -m\right)$$
where *f*(*x*) is the function of the original spectral curve, *h* is the increment of the independent variable, *f*′(*x*) is the FD, *f*″(*x*) is the SD, *f*^*α*^(*x*) is the *α*-order derivative, *α* can be any real number, Γ() is the gamma function, and *t* is the number of fitting points used in calculating the FOD, which is 40 in this paper.

It is worth noting that in this study, FD and SD were calculated by the 3-point binomial fitting curve. The FOD was calculated by the G-L method, which is defined based on discrete points. Therefore, the FD and SD calculated by this method may be slightly different from the results obtained by using the 3-point binomial fitting curve [38].

#### 2.3.2. Modeling Method and Model Evaluation

PLSR is a modeling method that integrates the advantages of principal component analysis, canonical correlation analysis, and multiple linear regression analysis. It is suitable for solving the modeling problem of variables with high collinearity and is considered to be a better linear modeling method in hyperspectral data modeling [39]. In its modeling process, it is necessary to select the number of potential variables. This study used the leave one cross-validation method to determine the number of latent variables [40].

This study used the sample subset partition based on the joint X–Y distances (SPXY) method to divide the 96 samples into a calibration set and a validation set according to the ratio of 2:1, including 64 samples in the calibration set and 32 samples in the validation set. The determination coefficient (R^2^), root mean square error (RMSE), and relative analysis error (RPD) were used to evaluate the accuracy of the model. Generally speaking, the closer R^2^ is to 1, the smaller the RMSE and the larger the RPD, indicating the higher accuracy of the model.

### 2.4. Software

In this paper, View Spec Pro 6.0 (ASD Company, Boulder, CO, USA) was used to eliminate abnormal values of the hyperspectral data; Microsoft Excel 2021 (Microsoft Corporation, Redmond, Washington, DC, USA), Unscrambler 9.7 (CAMO Software AS., Trondheim, Norway), and MATLAB 2021(MathWorks. Inc., Natick, MA, USA) were used to organize the data and construct the model, and Origin 2021(OriginLab Corporation, Northampton, MA, USA) was used for mapping.

## 3. Results

### 3.1. Data Characteristics of Total Flavonoids and Total Phenols in Tartary Buckwheat Grains

The maximum and minimum values of the total set of total flavonoids and total phenols were assigned to the calibration set (Table 2). The average value, standard deviation, and coefficient of variation in the total set, calibration set, and validation set of total flavonoids content were close. The average value, standard deviation, and coefficient of variation in the validation set of total phenols content were smaller than those of the total set and the calibration set. In general, the data distribution was reasonable and can be used for subsequent analysis.

### 3.2. Effect of Preprocessing Algorithm on Spectral Reflectance

In this study, two samples close to the average content of total flavonoids and total phenols in two data sets were taken as examples to analyze the changes in the original spectrum and its transformation spectra. It can be seen from the Figure 1 that the change trends of the original spectral reflectance of the two samples are basically the same. Both of them basically showed a trend of increase first and then decrease with the increase in wavelength, and both of them formed multiple reflection peaks and absorption valleys. Among the five transformation spectra, except MSC, the other four algorithms all had a great change effect on the original spectrum. The SNV algorithm did not change the overall change trend of the original spectral curve but changed the value of spectral reflectance. The change trend of the Lg spectrum was opposite to that of the original spectral curve. The values of the FD and SD spectra were mainly positive and negative alternately, and the changes in the SD spectrum were more frequent. These two algorithms made the original spectrum lose all its original spectral features. However, MSC transformation had no visible difference from the original spectral reflectance. It can be seen that different preprocessing algorithms had different effects on the changes in the original spectral curve.

### 3.3. Correlation Analysis between Transformation Spectra and Total Flavonoids and Total Phenols in Grains

Figure 2 shows the correlation between the original spectrum and different transformation spectra and the content of total flavonoids and total phenols in grains. The correlation of total flavonoids content and total phenols content with the same transformation spectrum was basically similar, but there were also some differences. The total flavonoids content and total phenols content were negatively correlated with the original spectrum, but positively correlated with the Lg spectrum. The two indicators were mainly negatively correlated with MSC and SNV spectra in the range of 401–800 nm, while they were mainly positively correlated in the range of 800–2450 nm. The correlation between the two indicators and FD and SD spectra was mainly positive and negative correlation alternating in the continuous band range, and the alternating phenomenon of the SD spectrum was more obvious. In addition, compared with the original spectrum, the correlation between the total flavonoids content and total phenols content and the transformation spectra was improved to a certain extent.

### 3.4. PLSR Monitoring Model of Total Flavonoids and Total Phenols in Grains

When constructing the PLSR model, it is necessary to determine the number of the best latent variables. This study used the minimum RMSE_cv_ as the standard to select the number of latent variables. Table 3 shows RMSE_cv_ and the number of latent variables of the monitoring model of total flavonoids and total phenols in grains based on different transformation spectra. The optimal number of latent variables of the total flavonoids content model based on R, MSC, SNV, Lg, FD, and SD was 11, 10, 11, 11, 5, and 3, respectively. Moreover, the optimal number of latent variables of the total phenols content model was 3, 1, 1, 3, 4, and 1, respectively.

Figure 3 shows the accuracy of the model under the optimal number of latent variables. The calibration accuracy of the total flavonoids content model constructed based on different transformation spectra was not much different, with R^2^_c_ distributed around 0.8 and RMSE_c_ distributed around 0.15, and the calibration accuracy under SNV transformation was the highest. The calibration accuracy of the total phenols content model based on different transformation spectra was different, and the calibration accuracy reached the highest under FD transformation. Different from the calibration accuracy, the total flavonoids content model achieved the best validation accuracy under SD transformation, and its RPD was 2.1462. The total phenols content model achieved the best validation accuracy under FD transformation, and its RPD was 1.8001. It can be seen that the total phenols content model obtained the best model under FD transformation. The model of total flavonoids content reached the highest validation accuracy in SD transformation, but its calibration accuracy also had a high level at this time, so SD can be used as the best preprocessing algorithm of total flavonoids content. It can be concluded that when the total flavonoids content and total phenols content models reached the best accuracy, the preprocessing algorithm used was derivative preprocessing. However, the FD and SD preprocessing cannot reflect the process of the original spectrum changing to the IOD spectrum. Therefore, in the following, this study will continue to study the effect of FOD preprocessing on the accuracy of the total flavonoids content and total phenols content model.

### 3.5. Hyperspectral Monitoring of Total Flavonoids and Total Phenols Based on FOD

#### 3.5.1. Effect of FOD on Spectral Reflectance

Figure 4 shows the spectral reflectance of the original spectrum (0 order) and 0.1~2 order (in steps of 0.1) FOD preprocessing. With the increase of order, the reflectance characteristics of the original spectral curve gradually disappeared. At the same time, FOD can show the change process from the original spectrum to the IOD spectrum. The main performance was that the value of spectral reflectance decreased gradually from order 0 to order 1.2, while the value of spectral reflectance increased gradually from order 1.3 to order 2.

#### 3.5.2. Correlation Analysis between FOD Spectra and Total Flavonoids and Total Phenols in Grains

Figure 5 shows the correlation between FOD spectra and the content of total flavonoids and total phenols in grains. In the range of order from 0 to 0.9, the FOD spectra was mainly negatively correlated with the content of total flavonoids and total phenols, while in the range of order from 1 to 1.5, it was mainly positively correlated. In the range of order from 0.9 to 2, there was a trend of alternating positive and negative correlation, and this phenomenon was more obvious with the increase of order. In addition, the FOD can show the gradual process of the correlation between the FOD spectra and the content of total flavonoids and total phenols at the same wavelength.

#### 3.5.3. PLSR Monitoring Model of Total Flavonoids and Total Phenols in Grains Based on FOD

Figure 6 shows the accuracy of the monitoring model of total flavonoids and total phenols in grains based on FOD. The R^2^_c_ of the total flavonoids content model showed little difference under different orders and reached the highest calibration accuracy at order 1.7, with R^2^_c_ and RMSE_c_ of 0.8879 and 0.1252, respectively. However, the calibration accuracy of the total phenols content model varied greatly under different orders, with the lowest accuracy at order 2 and the highest accuracy at order 1.8. The validation accuracy of the total flavonoids content model basically showed that with the increase in order, the validation accuracy basically showed a gradually increasing trend, and reached the highest validation accuracy at an order of 1.6. The validation accuracy of the total phenols content model was similar to the calibration accuracy. Its validation accuracy varied greatly under different orders and reached the highest validation accuracy at order 0.6. At the same time, when each model reached the highest validation accuracy, its calibration accuracy was also high. It can be concluded that the best models of total flavonoids content and total phenols content can be obtained under FOD preprocessing of orders of 1.6 and 0.6, respectively. Among them, the R^2^_c_, RMSE_c_, R^2^_v_, RMSE_v_, and RPD of the best model of total flavonoids content were 0.8731, 0.1332, 0.8384, 0.1448, and 2.4475, respectively. The R^2^_c_, RMSE_c_, R^2^_v_, RMSE_v_, and RPD of the best model of total phenols content were 0.8296, 0.2025, 0.6535, 0.1740, and 1.6713, respectively. At the same time, these two models were also the best monitoring models of total flavonoids content and total phenols content obtained in this study.

## 4. Discussion

Spectroscopy believes that different chemical bonds such as C-O, O-H, and C-H have different spectral characteristics [41]. Different substances are composed of different molecular structures, so different substances can be guaranteed to have different spectral characteristics. This is also the physical principle of quantitative monitoring of substances using hyperspectral technology [42]. Flavonoid refers to a series of compounds formed by the interconnection of benzene rings with phenolic hydroxyl groups through the central three carbon atoms [43,44]. Phenol refers to substances with one or more hydroxyl groups on a benzene ring [45]. Therefore, the structures of flavonoid and phenol make it possible to be quantitatively monitored by hyperspectral technology.

Research has showed that proper preprocessing of hyperspectral data before constructing a hyperspectral monitoring model can effectively reduce noise interference and improve model accuracy [27]. It is generally believed that different preprocessing algorithms have different denoising effects [33,34,46,47]. For example, MSC and SNV can eliminate the scattering influence of the spectrum to varying degrees, Lg transformation can reduce the influence of multiplicative factors caused by changes in lighting conditions, and derivative transformation can amplify the spectral characteristics. In this study, before constructing the hyperspectral monitoring model, we first considered the use of MSC, SNV, Lg, FD, and SD, five preprocessing algorithms, in order to preprocess the original hyperspectral data. The results showed that, except for the MSC, the other four algorithms had a greater effect on the spectral reflectance than the original spectrum. This was similar to the previous research results on other indicators, which was mainly determined by the nature of these algorithms [48,49,50]. After preprocessing of the original spectra, the correlation between the transformation spectra and the contents of total flavonoids and total phenols in grains was further analyzed. The results showed that the correlation change trend between total flavonoids content and transformation spectra was similar to that of total phenols content, and there was only a certain difference in the correlation coefficient at the same wavelength. In addition, compared with the original spectrum, each transformation spectrum improved the correlation between the spectral data and the content of total flavonoids and total phenols in grains to varying degrees. Yang et al. [50] carried out MSC, SNV, Lg, and FD preprocessing on the original spectrum before using hyperspectral technology to monitor the urease activity. The results showed that the preprocessing algorithm changed the spectral curve to varying degrees and improved the correlation between spectral reflectance and urease activity. The results of this study were similar. It can be seen that each preprocessing algorithm can reduce the spectral noise and improve the sensitivity of spectral information to varying degrees. In this study, PLSR was used to construct monitoring models for the content of total flavonoids and total phenols in grains. Combined with the previous experience [19,40], the cross-validation method was used to determine the optimal number of latent variables, and the model with the optimal number of latent variables was the best model under the preprocessing. The results showed that the calibration accuracy and validation accuracy of the total flavonoids content monitoring model based on different preprocessing algorithms were similar, and the best model was obtained under SD transformation. However, the calibration accuracy and validation accuracy of the total phenols content monitoring model based on different preprocessing algorithms were different to varying degrees. Among them, the SD transformation was the worst, while the FD transformation was the best. It can be seen that the accuracy of the model constructed based on different preprocessing algorithms was different. Yang et al. [25] constructed the rutin content models based on SD and constant offset elimination spectra, and believed that the model based on constant offset elimination spectra was more accurate. This was different from the results of this study, which may be caused by different data and preprocessing algorithms. Therefore, it is necessary to properly preprocess the hyperspectral data before constructing the hyperspectral monitoring model.

The above research results showed that the monitoring models of total flavonoids and total phenols achieved the best monitoring accuracy in derivative preprocessing. Therefore, this study raised the question of whether using FOD to preprocess the spectrum can obtain a higher accuracy. Based on this, this study further used the FOD algorithm to preprocess hyperspectral data. The results showed that FOD preprocessing can show the change process from the original spectrum to the IOD spectrum, which was mainly manifested in the gradual decrease of spectral reflectance in the range of order from 0 to 1.2 and the gradual increase in the range of order from 1.3 to 2. This was similar to the previous research results on other indicators [51]. The results of correlation analysis showed that at the same wavelength, with the change in order, the correlation between the preprocessing spectra and the content of total flavonoids and total phenols in grains did not have a certain law, but it can be seen that at the same wavelength, the correlation showed a gradual change trend with the change in order. In addition, in the range of order from 0 to 0.9, the FOD spectra was mainly negatively correlated with the content of total flavonoids and total phenols in grains, while in the range of order from 0.9 to 2, the positive and negative correlation appeared alternately. It can be seen that derivative preprocessing in the range of order from 0.9 to 2 played a more complex role in changing the correlation between spectral data and total flavonoids and total phenols in grains. The modeling results showed that the monitoring models of total flavonoids and total phenols reached the best accuracy under 1.6-order and 0.6-order preprocessing, respectively. At the same time, they were also the two best models of total flavonoids content and total phenols content in this study. The R^2^_c_, RMSE_c_, R^2^_v_, RMSE_v_, and RPD of the total flavonoids content model were 0.8731, 0.1332, 0.8384, 0.1448, and 2.4475, respectively. The R^2^_c_, RMSE_c_, R^2^_v_, RMSE_v_, and RPD of the total phenols content model were 0.8296, 0.2025, 0.6535, 0.1740, and 1.6713, respectively. The rapid monitoring of total flavonoids and total phenols in tartary buckwheat grains can be achieved, respectively. Yang et al. [52] achieved the highest accuracy under FOD preprocessing when constructing the hyperspectral monitoring model of soil total nitrogen. This shows that when considering the effect of derivative preprocessing on the hyperspectral model, it is necessary to consider both IOD and FOD. It is generally believed that the FD and SD represent the slope and curvature of the original spectral curve, respectively. FOD can reflect the sensitivity of the spectral curve slope and curvature changes [53]. Therefore, this may be one of the reasons why this study obtained the best monitoring model under FOD preprocessing.

## 5. Conclusions

In this study, tartary buckwheat was treated with selenium, and the total flavonoids content, total phenols content, and hyperspectral reflectance of tartary buckwheat grains were obtained. Moreover, the effect of different preprocessing algorithms on the accuracy of the PLSR monitoring model of total flavonoids content and total phenols content in tartary buckwheat grains was studied. The main conclusions were as follows: SNV, Lg, FD, and SD transformations changed the original spectral curve to varying degrees. Compared with the original spectrum, MSC, SNV, Lg, FD, and SD transformations spectra all improved the correlation between spectral data and the content of total flavonoids and total phenols in grains to varying degrees. The PLSR models of total flavonoids and total phenols in grains based on MSC, SNV, Lg, FD, and SD transformations spectra achieved the best accuracy under SD and FD transformation, respectively. FOD can show the change process from the original spectrum to the IOD spectrum, as well as the gradual process of the correlation between the transformation spectra at the same wavelength and the content of total flavonoids and total phenols. The PLSR monitoring model of total flavonoids and total phenols content in grains based on FOD achieved the best accuracy under the 1.6-order and 0.6-order derivatives, respectively. This study can provide a theory for realizing hyperspectral monitoring of total flavonoids and total phenols in tartary buckwheat grains. In addition, in future research, more preprocessing algorithms and modeling algorithms should be considered to construct models capable of obtaining more accurate results.

## Figures and Tables

**Figure 1 foods-12-01354-f001:**
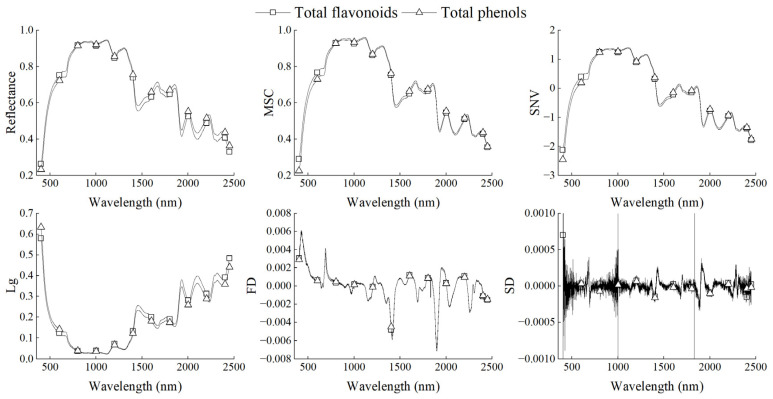
Spectral reflectance of different preprocessing.

**Figure 2 foods-12-01354-f002:**
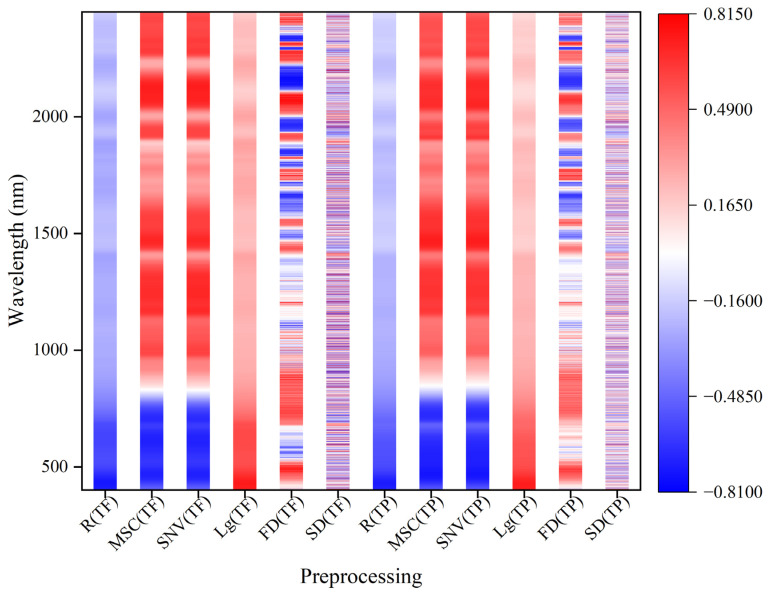
Correlation between transformation spectra and total flavonoids and total phenols in grains. R, TF, and TP mean the original spectrum, total flavonoids, and total phenols; the same applies below.

**Figure 3 foods-12-01354-f003:**
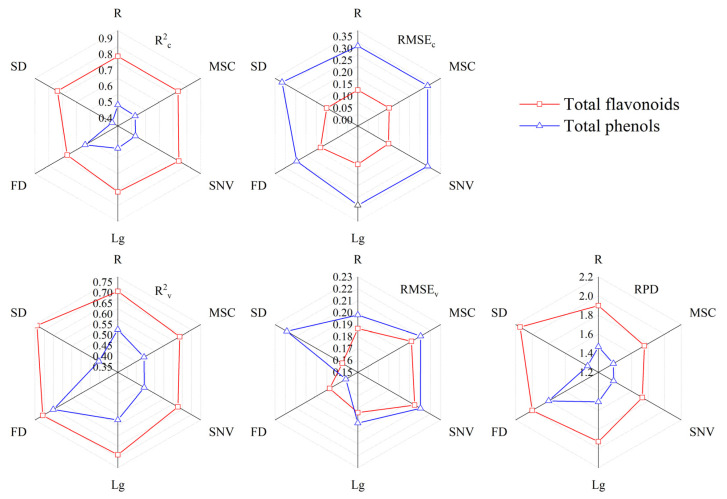
PLSR monitoring model of total flavonoids and total phenols in grains.

**Figure 4 foods-12-01354-f004:**
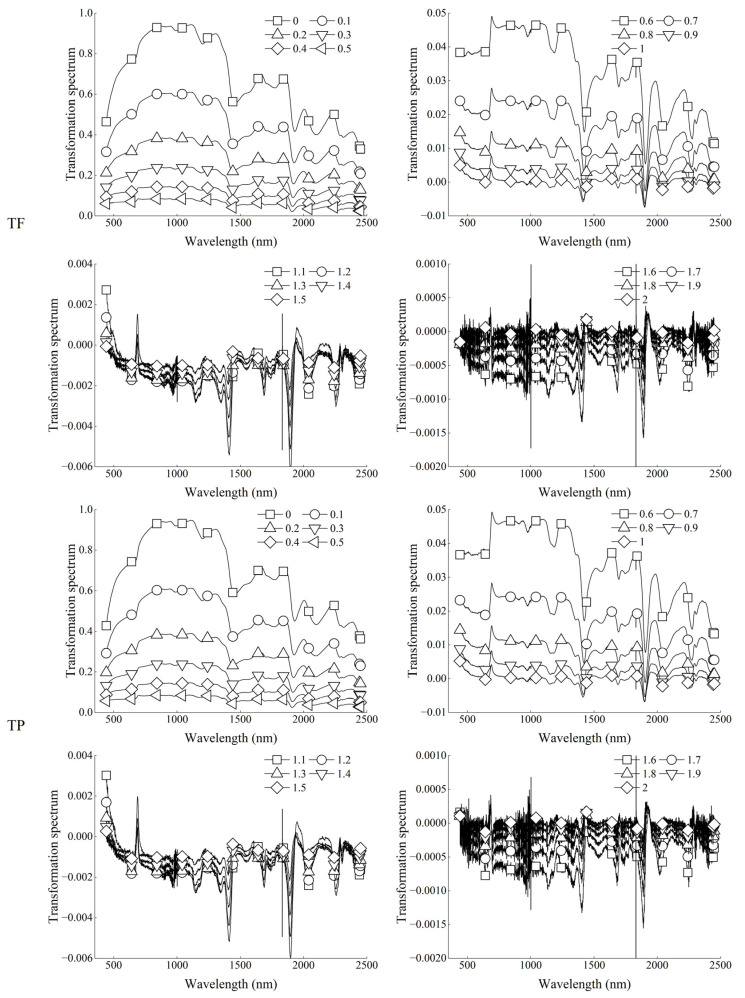
Spectral reflectance after FOD preprocessing.

**Figure 5 foods-12-01354-f005:**
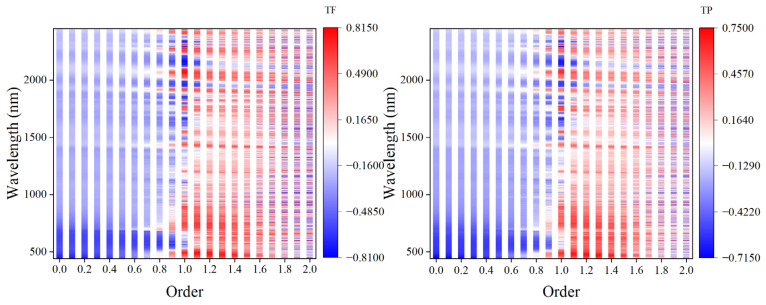
Correlation between FOD spectra and total flavonoids and total phenols in grains.

**Figure 6 foods-12-01354-f006:**
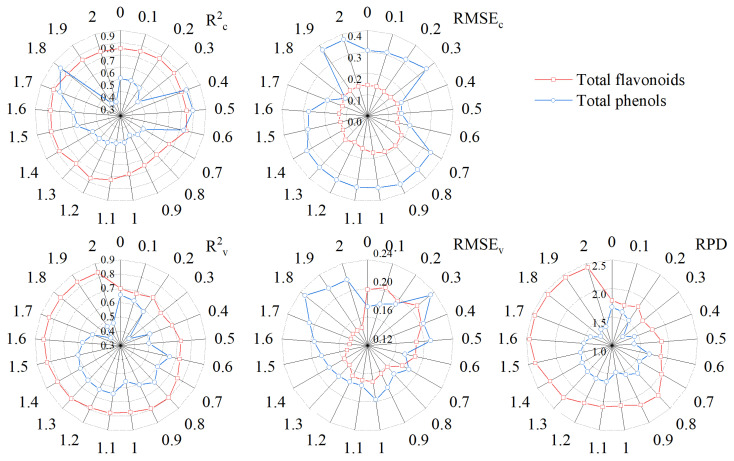
PLSR monitoring model of total flavonoids and total phenols in grains based on FOD.

**Table 1 foods-12-01354-t001:** L16 (8 × 23) orthogonal test scheme.

Number	Concentration (g·hm^−2^)	Varieties	Selenium Application Stage	Selenium Application Method
1	C1	V1	S1	M1
2	C1	V2	S2	M2
3	C2	V1	S1	M1
4	C2	V2	S2	M2
5	C3	V1	S1	M2
6	C3	V2	S2	M1
7	C4	V1	S1	M2
8	C4	V2	S2	M1
9	C5	V1	S2	M1
10	C5	V2	S1	M2
11	C6	V1	S2	M1
12	C6	V2	S1	M2
13	C7	V1	S2	M2
14	C7	V2	S1	M1
15	C8	V1	S2	M2
16	C8	V2	S1	M1

Note: C1, C2, C3, C4, C5, C6, C7, and C8 represent 8 selenium application concentrations of 0, 1.37, 2.74, 5.48, 8.22, 12.33, 18.495, and 27.7425 g·hm^−2^, respectively. V1 means that ‘Heifeng 1 hao’ is planted in both Taigu and Youyu, and V2 means that ‘Jinqiao 2 hao’ is planted in Taigu and ‘Jinqiao 6 hao’ is planted in Youyu. S1 means that the selenium application stages in Taigu and Youyu are the branching stage and early flowering stage, respectively. S2 means that the selenium application stages in Taigu and Youyu are the early flowering stage and peak flowering stage, respectively. M1 and M2 mean that the selenium application methods are soil selenium application and foliar selenium application, respectively.

**Table 2 foods-12-01354-t002:** Data characteristics of total flavonoids and total phenols.

	Data Sets	Num	Min (%)	Max (%)	Ave (%)	SD	CV
Total flavonoids	Total set	96	0.4275	1.9275	0.9733	0.3653	0.3754
	Calibration set	64	0.4275	1.9275	0.9985	0.3708	0.3714
	Validation set	32	0.4329	1.4507	0.9227	0.3543	0.3840
Total phenols	Total set	96	1.9115	4.0968	2.7298	0.4352	0.1594
	Calibration set	64	1.9115	4.0968	2.7787	0.4865	0.1751
	Validation set	32	2.0145	3.2471	2.6319	0.2909	0.1105

Notes: Num, Min, Max, Ave, SD, and CV mean the number, minimum, maximum, average, standard deviation, and coefficient of variation.

**Table 3 foods-12-01354-t003:** Selection of latent variables of the model based on different transformation spectra.

Lvs	Total Flavonoids						Total Phenols					
	R	MSC	SNV	Lg	FD	SD	R	MSC	SNV	Lg	FD	SD
1	0.3436	0.2773	0.2771	0.3310	0.2913	0.2253	0.4534	0.3516	0.3513	0.4351	0.4204	0.4016
2	0.2838	0.2761	0.2758	0.2825	0.2627	0.2115	0.3716	0.3561	0.3555	0.3630	0.3902	0.4116
3	0.2763	0.2745	0.2738	0.2769	0.2457	0.2098	0.3642	0.3583	0.3572	0.3598	0.3697	0.4179
4	0.2796	0.2502	0.2473	0.2794	0.2297	0.2101	0.3782	0.3690	0.3680	0.3707	0.3558	0.4149
5	0.2587	0.2497	0.2490	0.2760	0.2234	0.2138	0.3779	0.3821	0.3811	0.3722	0.3595	0.4265
6	0.2562	0.2512	0.2508	0.2708	0.2273	0.2211	0.4116	0.3958	0.3910	0.4053	0.3727	0.4185
7	0.2705	0.2524	0.2531	0.2753	0.2484	0.2302	0.3960	0.3923	0.3895	0.4175	0.3753	0.4298
8	0.2714	0.2345	0.2470	0.2744	0.2578	0.2397	0.4139	0.3861	0.3855	0.4124	0.3713	0.4321
9	0.2552	0.2344	0.2355	0.2703	0.2624	0.2376	0.4032	0.3811	0.3886	0.4129	0.3738	0.4302
10	0.2356	0.2291	0.2357	0.2530	0.2718	0.2353	0.3918	0.3775	0.3831	0.4085	0.3696	0.4316
11	0.2225	0.2297	0.2355	0.2523	0.2748	0.2347	0.3896	0.3793	0.3825	0.4168	0.3742	0.4328
12	0.2231	0.2308	0.2362	0.2588	0.2722	0.2346	0.3890	0.3995	0.3802	0.4189	0.3667	0.4322
13	0.2307	0.2424	0.2369	0.2610	0.2726	0.2334	0.3951	0.3970	0.4034	0.4226	0.3692	0.4323
14	0.2352	0.2545	0.2550	0.2650	0.2711	0.2333	0.4012	0.3870	0.4000	0.4363	0.3642	0.4326
15	0.2391	0.2601	0.2599	0.2778	0.2721	0.2329	0.3973	0.3789	0.3895	0.4306	0.3562	0.4328
16	0.2517	0.2587	0.2680	0.2879	0.2750	0.2328	0.3967	0.3734	0.3775	0.4268	0.3576	0.4326
17	0.2711	0.2503	0.2590	0.2929	0.2745	0.2326	0.3749	0.3731	0.3798	0.4221	0.3601	0.4327
18	0.2592	0.2499	0.2524	0.2822	0.2762	0.2326	0.3845	0.3862	0.3727	0.4260	0.3611	0.4326
19	0.2683	0.2534	0.2569	0.2806	0.2765	0.2326	0.3850	0.3932	0.3878	0.4220	0.3624	0.4326
20	0.2561	0.2580	0.2610	0.2856	0.2753	0.2326	0.4122	0.4104	0.3929	0.4254	0.3622	0.4325

Note: Lvs is the number of latent variables.

## Data Availability

The data presented in this study are available on request from the corresponding author.

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
