# Peer review of "Study on Hyperspectral Monitoring Model of Total Flavonoids and Total Phenols in Tartary Buckwheat Grains"

_foods, 2023, doi:10.3390/foods12071354_

Round 1

Reviewer 1 Report

The manuscript is written with clear understanding of the project addressed. However, there are major concerns that need to be addressed to enhance the quality of the manuscript. My specific comments are as follows:

Abstract: L25: “ Their R2c,RMSEc,R2v,RMSEv, and RPD were 0.8731, 0.1332, 0.8384, 0.1448, 2.4475, and 0.8296, 0.2025, 0.6535, 0.1740, 1.6713.” Which results specify for which flavonoids/total phenols?

Introduction:

Based on your objectives, please compare how your study is different from those that have already been published.

Materials and Methods:

Table 1: 1 and 2 represented what value/type/indicator? Please specify in the footnotes

L103: ‘FieldSpec 3 portable hyperspectral radiometer..” Add brand, state, country. Check for other equipment as well.

How about storage/treatment information of buckwheat?

How many split ratio for calibration/validation dataset?

There is no information regarding the data analysis of the study. Please add.

Results and Discussion:

Figure 1: Explain in details the application of using different pre-processing for the spectral reflectance.

Figure 3: it is hard to comprehend the differences between flavonoids/phenols. It could be because of the symbol used to represent the indicator. Instead, you could transforms it into table for easy reading

Figure 5: too small. Select only the main figures to represent your results

Relate your results with existing literatures to support your findings. Instead of mentioning the results, the authors should justify/explain the findings

General comments:

Please check the reference styles and grammar of the manuscript.

Author Response

Dear Reviewer:

Thank you for your valuable comments on my manuscript to make my manuscript better. The following is a detailed description of my modification of the manuscript according to your modification comments.

The manuscript is written with clear understanding of the project addressed. However, there are major concerns that need to be addressed to enhance the quality of the manuscript. My specific comments are as follows:

Abstract: L25: “ Their R2c,RMSEc,R2v,RMSEv, and RPD were 0.8731, 0.1332, 0.8384, 0.1448, 2.4475, and 0.8296, 0.2025, 0.6535, 0.1740, 1.6713.” Which results specify for which flavonoids/total phenols?

Modified. The original sentence has been modified and is located in lines 25-27.

Introduction:

Based on your objectives, please compare how your study is different from those that have already been published.

Modified. Add relevant content in lines 74-78, and combine the content in lines 62-63 of the original text, which is the main difference between this research and the published research.

Materials and Methods:

Table 1: 1 and 2 represented what value/type/indicator? Please specify in the footnotes

Modified. Table 1 has been modified and footnotes have been added to make the information in Table 1 easier to understand. Relevant contents are in lines 107-115.

L103: ‘FieldSpec 3 portable hyperspectral radiometer..” Add brand, state, country. Check for other equipment as well.

Modified. Brands, states, and countries are added to the instruments and software used in this paper. Relevant contents are in lines 120, 125 and 184-188.

How about storage/treatment information of buckwheat?

Modified. Add detailed information about collecting buckwheat samples and obtaining grain powder. The relevant contents are in lines 117-119.

How many split ratio for calibration/validation dataset?

Modified. The split ratio of calibration set and validation set is 2:1, and relevant contents are added in lines 177-178.

There is no information regarding the data analysis of the study. Please add.

Modified. We have added data analysis in lines 132-137 and made adjustments to the structure of section 2.3.

Results and Discussion:

Figure 1: Explain in details the application of using different pre-processing for the spectral reflectance.

Modified. The analysis of spectral change characteristics of different preprocessing is added. Relevant contents are in lines 220-224.

Figure 3: it is hard to comprehend the differences between flavonoids/phenols. It could be because of the symbol used to represent the indicator. Instead, you could transforms it into table for easy reading

Modified. Change the original Figure 3 to Table 3.

Figure 5: too small. Select only the main figures to represent your results

Modified. Due to the large difference in the range of the y-axis between different figures, it is impossible to reduce the number of figures by deleting part of the curve and combining the two figures into one figure, because the change trend of some curves will not be seen. However, we enlarged the font size and size of the figure to make it clearer.

Relate your results with existing literatures to support your findings. Instead of mentioning the results, the authors should justify/explain the findings

Modified. In the Discussion, the results are compared with the existing literature. Relevant contents are in lines 373-377, 390-394, and 421-424.

General comments:

Please check the reference styles and grammar of the manuscript.

Modified. The reference style and grammar of the manuscript are checked and modified.

Reviewer 2 Report

The manuscript is adequately written and meets scientific standard. Appropriate design of the experiment and descriptions of measurements as well as the results and discussions have been provided. The authors applied statistical techniques to evaluate total flavonoids and phenols in buckwheat grains. The English language is satisfactory, and readers will understand the information presented. However, it is necessary that the authors revise their work according to the following comments:
General comment:
The manuscript is not properly organized according the the Journal's Instructions for Authors.
- The authors should follow the Instructions accordingly - from the Title until References.
- The format/style of every information should be followed closely.
Specific Comments
- Check the format for writing the Authors names with superscript numbers. Define their affiliations appropriately. Line 7 information is not appropriate.  Refer to the Journal's Instructions, please.
Objectives:
- Define or state the objectives clearly. This paper aims to: do not agree with the 'If' and 'It' in objectives (2) and (3). Revise the objectives accordingly for clarity.
- The Equation numbers (1) to (4) should be moved to the right side.
- Check the format/style for lines 87, 100, 110, 111, 142, 170, 185, 203, 236, 237, and 260.
- Section 3.5.3 should not begin with the Fig. 7. Provide the explain first and the figure below it. This should also be corrected in Sections 3.2 to 3.5. Ensure that the 'Results' Sections do not begin the Figures. It is not scientifically appropriate. Refer to other published papers in Foods or Instructions for Authors.
Tables and Figures:
- Table 1 and Table 2 should be bolded.
- Fig. should be Figure, and should be bolded.
- Refer to other published papers in Foods or Instructions for Authors.
Conclusions.
- Ensure that the conclusions becomes a single paragraph. Remove the numbers. Add also the values statement or future directions for further studies.
References
- Ensure that the references are in accordance with the Journal's Instructions for authors. Refer to it, please.

Author Response

Dear Reviewer:

Thank you for your valuable comments on my manuscript to make my manuscript better. The following is a detailed description of my modification of the manuscript according to your modification comments.

The manuscript is adequately written and meets scientific standard. Appropriate design of the experiment and descriptions of measurements as well as the results and discussions have been provided. The authors applied statistical techniques to evaluate total flavonoids and phenols in buckwheat grains. The English language is satisfactory, and readers will understand the information presented. However, it is necessary that the authors revise their work according to the following comments:

General comment:

The manuscript is not properly organized according the the Journal's Instructions for Authors.

- The authors should follow the Instructions accordingly - from the Title until References.

- The format/style of every information should be followed closely.

Modified. The full text format has been modified according to the format requirements of Foods.

Specific Comments
- Check the format for writing the Authors names with superscript numbers. Define their affiliations appropriately. Line 7 information is not appropriate.  Refer to the Journal's Instructions, please.

Modified. The wrong superscript and line 7 information are deleted.Objectives:
- Define or state the objectives clearly. This paper aims to: do not agree with the 'If' and 'It' in objectives (2) and (3). Revise the objectives accordingly for clarity.

Modified. The objective (2) is rewritten and the objective (3) is modified. The relevant contents are in lines 87-92.- The Equation numbers (1) to (4) should be moved to the right side.

Modified. The number is placed on the right. Relevant contents are in lines 148 and 158-160.- Check the format/style for lines 87, 100, 110, 111, 142, 170, 185, 203, 236, 237, and 260.

Modified. The format of relevant content has been modified according to the format requirements of the journal. The relevant contents are in lines 94, 116, 131, 138, 169, 183, 191, 209, 230-231, 252, 290-291, 303-304, and 318-319.- Section 3.5.3 should not begin with the Fig. 7. Provide the explain first and the figure below it. This should also be corrected in Sections 3.2 to 3.5. Ensure that the 'Results' Sections do not begin the Figures. It is not scientifically appropriate. Refer to other published papers in Foods or Instructions for Authors.

Modified. The Figures and Tables in the ‘Results’ section are placed after the text.Tables and Figures:
- Table 1 and Table 2 should be bolded.

- Fig. should be Figure, and should be bolded.
- Refer to other published papers in Foods or Instructions for Authors.

Modified. The caption format of the Figures and Tables is modified according to the requirements of Foods. The relevant contents are in lines 107, 199, 229, 249-251, 264, 289, 302, 317, and 343.Conclusions.
- Ensure that the conclusions becomes a single paragraph. Remove the numbers. Add also the values statement or future directions for further studies.

Modified. Change the conclusion into a paragraph, and add the research value and future research direction of this study. The relevant contents are in lines 430-451.References
- Ensure that the references are in accordance with the Journal's Instructions for authors. Refer to it, please.

Modified. The format of references has been modified according to the requirements of Foods.

Round 2

Reviewer 1 Report

The authors have addressed all the comments.